# KaRF: Weakly-Supervised Kolmogorov-Arnold Networks-based Radiance Fields for Local Color Editing

**Wudi Chen**
Jilin University

**Zhiyuan Zha**[*]
Jilin University

**Shigang Wang**
Jilin University

**Bihan Wen**
Nanyang Technological University

**Xin Yuan**
Westlake University

**Jiantao Zhou**
University of Macau

**Zipei Fan**
Jilin University

**Gang Yan**
Jilin University

**Ce Zhu**
University of Electronic Science and Technology of China

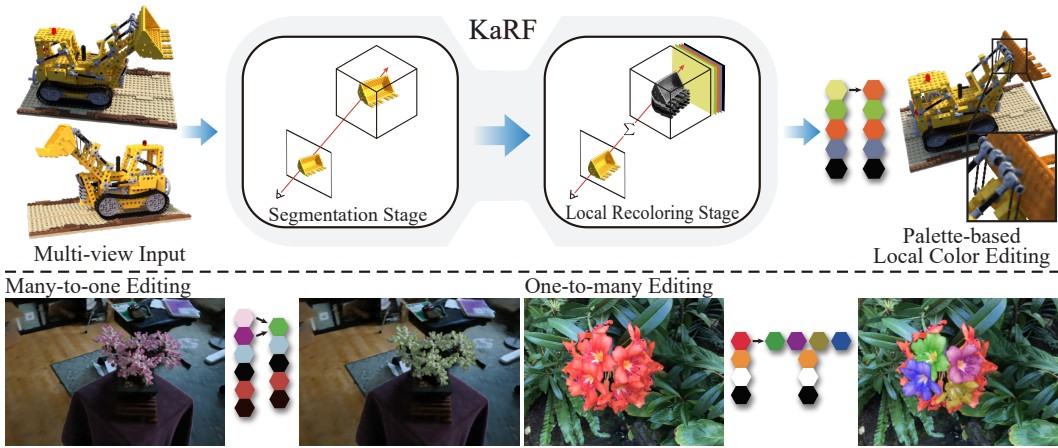

Figure 1: We propose KaRF, a novel weakly-supervised method for local color editing of neural radiance fields. **KaRF** constructs a two-stage **K**olmogorov-**a**rnold Networks-based **R**adiance **F**ields, which achieves precise region segmentation and efficient local recoloring tasks. Our method enables accurate and natural local color editing of arbitrary regions within 3D scenes, while also supporting one-to-many or many-to-one editing mappings, greatly enhancing the flexibility and expressiveness of local color editing.

## Abstract

Recent advancements have suggested that neural radiance fields (NeRFs) show great potential in color editing within the 3D domain. However, most existing NeRF-based editing methods continue to face significant challenges in local region editing, which usually lead to imprecise local object boundaries, difficulties in maintaining multi-view consistency, and over-reliance on annotated data. To address these limitations, in this paper, we propose a novel weakly-supervised method called KaRF for local color editing, which facilitates high-fidelity and realistic appearance edits in arbitrary regions of 3D scenes. At the core of the proposed KaRF approach is a unified two-stage Kolmogorov-Arnold Networks (KANs)-based radiance fields framework, comprising a segmentation stage followed by a local recoloring stage. This architecture seamlessly integrates geometric priors from NeRF to achieve weakly-supervised learning, leading to superior performance. More specifically, we propose a residual adaptive gating KAN structure, which integrates KAN with residual connections, adaptive parameters, and gating mechanisms to effectively enhance segmentation accuracy and refine specific

---

[*]Corresponding author

39th Conference on Neural Information Processing Systems (NeurIPS 2025).

editing effects. Additionally, we propose a palette-adaptive reconstruction loss, which can enhance the accuracy of additive mixing results. Extensive experiments demonstrate that the proposed KaRF algorithm significantly outperforms many state-of-the-art methods both qualitatively and quantitatively. Our code and more results are available at: https://github.com/PaiDii/KARF.git.

# 1 Introduction

Neural radiance fields (NeRFs) [29] are capable of constructing high-quality 3D scenes and rendering fine-grained and photorealistic novel views by fusing 2D multi-view images with camera pose information. With its implicit representation-driven 3D scene modeling capability, NeRF has been successfully applied in various domains [16, 13, 37] and scenes [31, 2, 38]. Nevertheless, the effective utilization of the 3D scene neural representation learned by NeRF to achieve precise local editing while maintaining a high degree of realism remains a critical frontier for further exploration. This challenge arises from the fully connected architecture used by NeRF, where adjusting a single parameter triggers global changes across all parameters, making the precise extraction of local regions in scene space extremely complex. Meanwhile, under the constraints of cross-view consistency, performing detailed and natural edits is also subject to significant limitations.

To address the challenges of NeRF local color editing, one class of methods focuses on image segmentation models, achieving local editing by distilling 2D semantic features into 3D feature fields [22, 20]. However, this category of methods often leads to color bleeding in non-edited regions. Other methods acquire local regions through point cloud projection [23] or voxel expansion followed by point cloud extraction [32, 17]. Due to the discrete nature of point clouds and the fixed volume of voxels, these methods have limited ability to accurately delineate the contours of local objects. Additionally, these methods struggle to distinguish between the approximate colors of objects, which limits their potential applications in the field of local editing.

Bearing the above concerns in mind, in this paper, we propose KaRF, a novel weakly-supervised method for local color editing by leveraging the efficient approximation ability and outstanding nonlinear expressive power of Kolmogorov-Arnold Networks (KANs) [26]. To the best of our knowledge, this is the first work to apply KAN to NeRF-based realistic editing for arbitrary regions. Using only one or three coarse, automatically generated masks per scene with no manual refinement, the proposed KaRF approach achieves consistent and precise local color editing effects, as shown in Figure 1. Specifically, we model local color editing as a unified two-stage process comprising segmentation and local recoloring. To this end, we propose a residual adaptive gating KAN structure that accurately segments arbitrary regions and performs localized optimization in accordance with user-specified color requirements. By incorporating geometric neural representations derived from the pre-trained NeRF, our proposed KaRF approach effectively addresses inconsistencies, low quality, and under-sampling masks or additive mixing weights from multi-view inputs. To further enhance model flexibility and training stability, we introduce multidimensional adaptive parameters. Based on this structure, we build radiance field models tailored to each stage and design a palette-adaptive reconstruction loss during the local recoloring stage to ensure a coherent distribution of the base colors in 3D space. Finally, KaRF forms an integrated, high-quality editing approach that seamlessly combines segmentation and local recoloring. The significant contributions of this paper are summarized as follows:

(1) We propose the KaRF framework for local color editing, which enables users to selectively recolor arbitrary regions while requiring minimal guiding information.

(2) We propose a novel residual adaptive gating KAN structure and a palette-adaptive reconstruction loss to achieve precise segmentation and local recoloring.

(3) Extensive experiments demonstrate that the proposed KaRF algorithm significantly outperforms many state-of-the-art methods both qualitatively and quantitatively.

# 2 Related Work

## 2.1 Segmentation in NeRF

NeRF segmentation tasks aim to generate high-precision and multi-view consistent segmentation masks, which are usually classified into two categories: feature-alignment-based methods [12, 18, 20]

and 2D model-based methods [30, 7]. Feature-alignment-based methods constructed view-consistent segmentation clusters by aligning 2D visual features with additional features fields. For instance, Semantic-NeRF [39] extended sparse semantic labels into dense semantic annotations by jointly encoding geometric, appearance, and semantic information. NeRF-SOS [12] integrated appearance features and geometric information into the segmentation field to produce segmentation masks. Conversely, 2D model-based approaches leveraged existing image segmentation priors to guide and constrain the generation of 3D segmentation masks. For example, the MVSeg module in SPIn-NeRF [30] and SA3D [7] generated multi-view consistent segmentation masks based on foundational 2D segmentation models [6, 19]. However, SPIn-NeRF suffered from noisy outputs due to insufficient integration of 3D information, whereas SA3D tended to generate segmentation masks with blurred boundaries under conditions of significant viewpoint variation. Compared to the aforementioned 2D model-based approaches, our approach exhibits superior performance in terms of both segmentation accuracy and consistency.

## 2.2 Color Editing in NeRF

Color editing tasks aim to modify the visual attributes of objects (*e.g.*, color, hue, and brightness) while maintaining photorealistic fidelity. Previous works [33, 3, 4] combined NeRF with physics-based rendering to explicitly decompose properties such as reflectance, geometry, and lighting, enabling realistic scene reconstruction and flexible control under varying lighting conditions. On the other hand, Upst-NeRF [10] and SttcNeRF [9] focused on optimizing NeRF, concentrating mainly on preserving geometric details and texture consistency of the scene under stylization constraints. However, recent advanced approaches, such as PaletteNeRF [22, 35] and RecolorNeRF [14] applied palette-based color editing to NeRF to achieve multi-view consistent and realistic effects. By optimizing palettes and decomposing color layers, enabling intuitive recoloring and high-fidelity editing of complex scenes. To further extend local editing capabilities, ICENeRF [23] enabled local recoloring in NeRFs by decoupling the perceptual optimization of color MLP weights. However, this method can struggle with precise local color control, and its foreground mask requirement scales with the number of colors being edited. IReNe [27] improves local editing speed by fine-tuning the last layer of its color MLP to learn local colors. Nevertheless, it can suffer from color bleeding in occluded situations and requires precise manual maps and new reference images for multiple edits. LAENeRF [32], through color layers decomposition based on ray termination points and voxel-based region expansion, achieved realistic editing of local regions. However, voxel expansion may result in imprecise region selection, which may affect subsequent appearance editing. In contrast, KaRF demonstrates superior localization and multi-view consistency in color editing.

## 3 Preliminary

**KAN.** Multilayer perceptrons (MLPs) [15] have become the cornerstone of numerous modern neural networks. Recently, Kolmogorov-Arnold Networks (KANs) [26] have been introduced as an alternative to MLPs. They are based on the Kolmogorov-Arnold representation theorem [21], which asserts that a sum of compositions of several univariate functions can represent any multivariate continuous function. Consequently, both MLPs and KANs can be regarded as models that utilize functional compositions and combinations to express complex mappings. In MLPs, however, the functions are represented by fixed activation functions applied to nodes, while the combinations are achieved through linear weight matrices connecting neurons. In contrast, the functions in KANs are learnable univariate functions on edges, while the nodes perform simple addition operations. The overall structure of an $N$-layer KAN is defined as:

$$\text{KAN}(\mathbf{X}) = (\mathbf{\Phi}_{N-1} \circ \mathbf{\Phi}_{N-2} \circ \cdots \circ \mathbf{\Phi}_1 \circ \mathbf{\Phi}_0)\mathbf{X}, \tag{1}$$

where $\mathbf{\Phi}_l$ denotes the $l$-th layer within the KAN model and is composed of 1D learnable activation functions ($\phi$):

$$\mathbf{\Phi} = \{\phi_{j,i}\} \quad i = 1, 2, \ldots n_{in}, \quad j = 1, 2, \ldots n_{out}, \tag{2}$$

where $n_{in}$ and $n_{out}$ are the number of input and output nodes in a single KAN layer, respectively. The computation from layer $l$ to layer $l + 1$ in the KAN model can be represented in the following

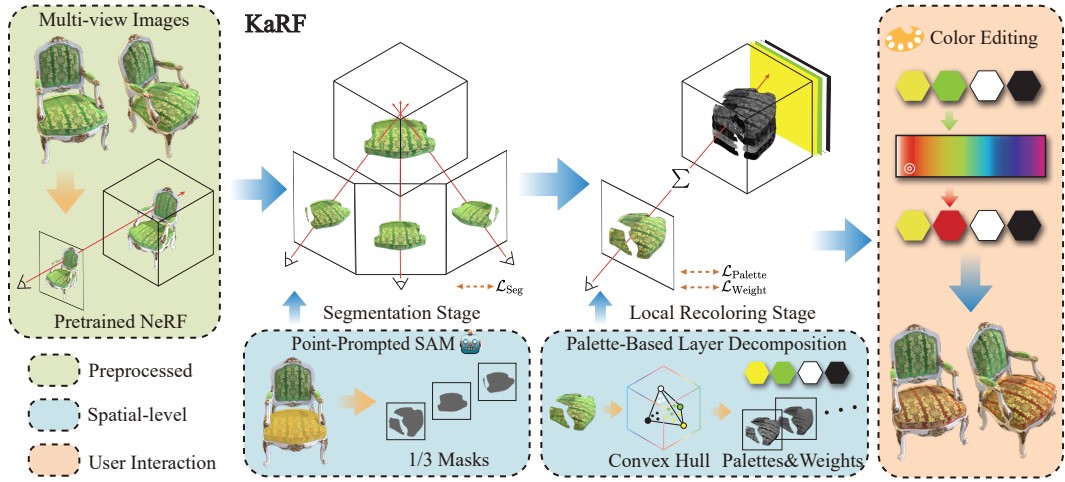

Figure 2: **Overview of KaRF.** We employ a pre-trained NeRF [29] to extract the output of its density MLP (*i.e.*, 3D point density $\sigma$ and multi-channel features $\boldsymbol{f}_{prior} \in \mathbb{R}^{256}$ containing scene prior knowledge). Subsequently, the NeRF parameters are frozen, and a two-stage training process is initiated: 1) Segmentation stage. Utilizing multi-view images and an initial set of 1-3 coarse segmentation masks generated by point-prompted SAM [19], we generate consistent masks by constructing a KAN-based segmentation radiance fields, which is supervised by $\mathcal{L}_{\text{Seg}}$. These refined masks are then passed to the next stage. 2) Local recoloring stage. We adopt the strategy from LoCoPalette [8] to compute convex hulls for the masks and perform layer decomposition, thereby obtaining initial palettes and weights. These parameters are then fed into a KAN-based recoloring radiance fields, aimed at the fine-grained reconstruction of weights in 3D space and the adaptive optimization of the palettes. This stage is supervised by $\mathcal{L}_{\text{Weight}}$ and $\mathcal{L}_{\text{Palette}}$. The trained palettes and weights enable users to interactively composite the scene by directly modifying colors in the palettes.

matrix form:

$$\mathbf{X}_{l+1} = \underbrace{\begin{bmatrix} \phi_{l,1,1}(\cdot) & \phi_{l,1,2}(\cdot) & \dots & \phi_{l,1,n_l}(\cdot) \\ \phi_{l,2,1}(\cdot) & \phi_{l,2,2}(\cdot) & \dots & \phi_{l,2,n_l}(\cdot) \\ \vdots & \vdots & \vdots & \vdots \\ \phi_{l,n_{l+1},1}(\cdot) & \phi_{l,n_{l+1},2}(\cdot) & \dots & \phi_{l,n_{l+1},n_l}(\cdot) \end{bmatrix}}_{\Phi_l} \mathbf{X}_l, \tag{3}$$

where $\mathbf{X}_l \in \mathbb{R}^{n_l}$ represents the input to layer $l$, and $\mathbf{X}_{l+1} \in \mathbb{R}^{n_{l+1}}$ is the resulting output of the same layer. $\phi(\cdot)$ denotes a learnable univariate function, parameterized as a B-spline curve [11] with learnable coefficients. Each $\phi_{l,j,i}(\cdot)$ models the relationship between the $j$-th output node and the $i$-th input node in layer $l$, enabling flexible, non-linear mappings between neurons. Mathematically, $\phi(\cdot)$ can be expressed as:

$$\phi(\cdot) = w_b b(\cdot) + w_s \xi(\cdot), \tag{4}$$

where $b(\cdot) = silu(\cdot)$ and $\xi(\cdot) = spline(\cdot) = \sum_k c_k B_k(\cdot)$. For more details on KAN, please refer to [26]. Due to the superior representation performance for complex nonlinear mappings, KANs can describe complex mappings with fewer parameters, demonstrating superior approximation capabilities and offering greater flexibility and efficiency compared to MLPs. Furthermore, the use of simple functional combinations makes the internal mechanisms of KANs more interpretable.

## 4 Methodology

In this section, we describe the details of KaRF (Sec. 4.1), followed by an introduction to the two-stage process of local color editing (Sec. 4.2) and the optimization strategies for each stage (Sec. 4.3). Figure 2 illustrates an overview of our two-stage pipeline.

### 4.1 KaRF

In NeRF [29], the inductive bias introduced through staged network design effectively constrains the solution space for complex function fitting, providing accurate geometric structure and appearance priors for NeRF-based downstream tasks (*e.g.*, segmentation, color editing), thereby enabling higher-quality outputs in post-processing. Building on this, KANs [26], with their exceptional nonlinear expressive power, are particularly well-suited for fitting complex mappings. Leveraging this strength

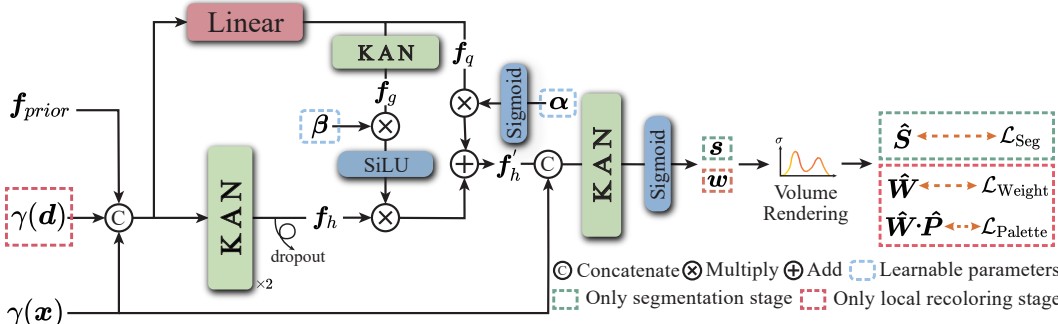

Figure 3: Our network structure. Our KAN-based segmentation radiance fields utilize positional encodings $\gamma(\boldsymbol{x})$ and prior features $\boldsymbol{f}_{prior}$ to output a per-point probability distribution $\boldsymbol{s} \in [0,1]^{N_k}$ over $N_k$ predefined classes. Subsequently, our KAN-based recoloring radiance fields then incorporate the viewing direction $\gamma(\boldsymbol{d})$ in addition to the aforementioned inputs, predicting a per-point weight vector $\boldsymbol{w} \in [0,1]^{N_p}$ corresponding to a palette $\hat{\boldsymbol{P}}$ containing $N_p$ colors. Both $\boldsymbol{s}$ and $\boldsymbol{w}$ are volume rendered to yield a per-pixel segmentation map $\hat{\boldsymbol{S}}$ and weight maps $\hat{\boldsymbol{W}}$.

and the priors from NeRF, we have constructed a versatile model centered around KAN, with its major components as follows:

• **GRBFKAN layers.** B-spline functions, owing to their piecewise polynomial nature, exhibit inherent advantages in modeling smooth functions. However, in scenarios involving high-frequency spatial details (*e.g.*, specular highlights or abrupt color transitions), such as local color editing, the representational accuracy of B-splines can be constrained. In view of this, we replace B-spline functions in vanilla KAN [26] with Gaussian Radial Basis Functions (GRBFs) [5]. GRBFs, utilizing Gaussian kernels, possess a key advantage in the adaptability of their scale parameters, enabling them to more effectively approximate color details characterized by drastic local changes or sharp features. Specifically, each KAN in KARF is composed of GRBFKAN layers, in which the response value of each GRBF within these layers depends on the Euclidean distance between the input and its respective center point $c_i$. This mechanism not only endows the model with excellent non-linear approximation capabilities and greater flexibility but also, in some cases, leads to improvements in computational performance [25]. Within this framework, $\xi(x)$ is represented as a GRBF function with $N$ centers as follows:

$$\xi(x) = \sum_{i=1}^{N} w_i \exp\left(-\frac{\|x - c_i\|^2}{2h^2}\right),\tag{5}$$

where $w_i$ denotes the trainable weight, and $h$ represents the bandwidth of the Gaussian kernel, which modulates the dispersion of the function and controls its response range.

• **Residual Adaptive Gating KAN.** Although stacking multiple layers of KAN can enhance the non-linear expressive capacity of the network, it leads to training instability and increases computational overhead. Consequently, we propose a KAN-based residual adaptive gating structure, illustrated in Figure 3. This structure, serving as a component designed for the segmentation and recoloring radiance fields, is capable of dynamically and finely adaptively regulating the feature flow.

Specifically, we first employ a computationally efficient two-layer KAN as the backbone network. This backbone is designed to capture critical geometric structures and appearance representations from the input data at a limited computational cost, thereby extracting the backbone non-linear features $\boldsymbol{f}_h$. Subsequently, to achieve fine-grained control over the feature flow, we designed a gating module. This module internally embeds a linear layer and a KAN layer, which are utilized to learn a context-dependent gating vector $\boldsymbol{f}_g$, for dynamically modulating the feature flow. However, considering the inherent heterogeneity among different feature channels, a singular $\boldsymbol{f}_g$ exhibits limitations in channel-specific regulation. To address this, we designed a trainable channel-wise adaptive operator $\boldsymbol{\beta} \in \mathbb{R}^{256}$. This operator performs element-wise selective activation and suppression on $\boldsymbol{f}_g$, thereby empowering the network with the capability to differentially process distinct semantic channels. Furthermore, shallow features $\boldsymbol{f}_q$, extracted by the linear layer within the gating module, are combined with a learnable modulation factor $\boldsymbol{\alpha} \in \mathbb{R}$ and injected into the main feature flow. This approach aims to enhance training stability and convergence speed. Ultimately, the update process for the backbone

features can be formalized by the following expression:

$$\boldsymbol{f}_h^{'} = \text{Sigmoid}(\boldsymbol{\alpha}) \cdot \boldsymbol{f}_q + \text{SiLU}(\boldsymbol{\beta}\boldsymbol{f}_g) \cdot \boldsymbol{f}_h, \tag{6}$$

where the enhanced backbone features $\boldsymbol{f}_h^{'}$ are then concatenated with positional encoding $\gamma(\boldsymbol{x})$, and fed into an output head, constructed from a single KAN layer, to produce the final segmentation mask or color weights.

## 4.2 Local Color Editing

**Segmentation stage.** During the geometric modeling stage of NeRF [29], density $\sigma$ and multi-channel features $\boldsymbol{f}_{prior}$ are generated to characterize the scene, with the latter encoding prior information about the structure and appearance of the scene. Building on this foundation, we propose fine-grained KAN-based segmentation radiance fields, which leverage inconsistent and geometrically coarse 2D segmentation masks (typically one for forward-facing scenes and three for 360° scenes) generated by Segment Anything Model (SAM) [19] to generate 3D segmentation results that exhibit refined boundary textures and multi-view consistency.

Specifically, we formalize scene segmentation as a view-invariant spatial mapping function. This function maps positional encodings $\gamma(\boldsymbol{x})$ and prior features $\boldsymbol{f}_{prior}$ (derived from a pre-trained NeRF) to a per-point probability distribution $\boldsymbol{s}$ using a softmax activation:

$$\text{softmax}\big(F_s(\gamma(\boldsymbol{x}), \boldsymbol{f}_{prior})\big) \rightarrow \boldsymbol{s}, \tag{7}$$

where $F_s$ represents our residual adaptive gating KAN in segmentation stage. The design of $F_s$ aims to ensure that the resulting segmentation exhibits consistent geometric structure.

Following this, $\boldsymbol{s}$ and $\sigma$ (obtained from a pre-trained NeRF) are integrated via overall fine volumetric rendering to assign a semantic label to each pixel in the segmentation map $\hat{\boldsymbol{S}}$:

$$\hat{\boldsymbol{S}} = \underset{k}{\text{argmax}} \left( \sum_{i=1}^{N_{\text{fine}}} T_i \left(1 - \exp(-\sigma_i \delta_i)\right) \boldsymbol{s}_i \right), \quad where \; T_i = \exp\left(-\sum_{j=1}^{i-1} \sigma_j \delta_j\right), \tag{8}$$

where $N_{\text{fine}}$ denotes the number of fine sampling points, $\delta_i$ is the distance between sample $i$ and sample $i+1$ and $k$ denotes a class index. Notably, our method supports multi-class weakly-supervised segmentation by simply expanding the channel dimension of the output head. This flexibility also broadens the applicability of our local color editing.

**Local Recoloring stage.** This stage comprises two essential steps: palette extraction and layer decomposition. In the palette extraction step, directly leveraging spatial-level color distribution for segmentation can result in global color inconsistencies from certain viewpoints, such as the abrupt appearance of red elements in a predominantly green scene. To address this, we first perform fine-grained segmentation of the scene and then extract the palette from the segmented objects, thereby ensuring spatial color consistency. The layer decomposition step aims to construct KAN-based recoloring radiance fields, which, through simple dimensional concatenation, can be decomposed into multi-dimensional radiance layers weighted by solid colors to enhance recoloring flexibility.

Unlike other 3D recoloring methods [22, 14, 32] that follow the strategy introduced by [34], we employ the LoCoPalette [8] approach to one or three segmented foreground objects to derive the palette $\boldsymbol{P} = \{p_1, p_2, ..., p_{N_P}\}$ and 2D sparse weight maps $\boldsymbol{W} = \{w_1, w_2, ..., w_{N_P}\}$. An output color $\boldsymbol{C}$ is composited from these components using the expression:

$$\boldsymbol{C} = \left\{ \sum_{i=0}^{N_P} w_i p_i \;\middle|\; w_i \in \boldsymbol{W} \text{ and } p_i \in \boldsymbol{P} \right\}, \tag{9}$$

where $N_P$ represents the number of base colors in $\boldsymbol{P}$. We leverage $\boldsymbol{W}$ as supervisory signals, enabling the generated results to effectively reduce the impact of primary color variations on non-primary colors. Additionally, background regions learn to select only a black base color, effectively isolating them from the recoloring process.

The local recoloring stage requires the viewing direction $\gamma(\boldsymbol{d})$ as an additional input, because lighting conditions and specular reflection intensity vary across different viewpoints. To this end, the recoloring radiance fields learn a mapping from $\gamma(\boldsymbol{x})$, $\gamma(\boldsymbol{d})$, and $\boldsymbol{f}_{prior}$ to a per-point weight vector $\boldsymbol{w}$:

$$F_w\big(\gamma(\boldsymbol{x}), \gamma(\boldsymbol{d}), \boldsymbol{f}_{prior}\big) \rightarrow \boldsymbol{w}, \tag{10}$$

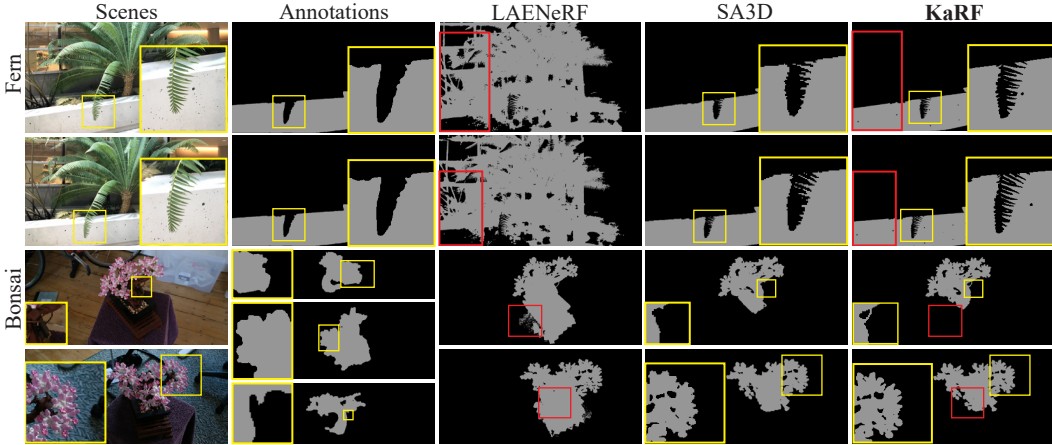

Figure 4: Qualitative comparison with NeRF segmentation methods. Zoom-in views are highlighted within the yellow boxes.

where $F_w$ represents our residual adaptive gating KAN in local recoloring stage.

Subsequently, we synthesize the RGB image by employing a volume rendering formula analogous to Eq. (8), along with Eq. (9). We observe that, although the global color scheme of local objects remains consistent, color inaccuracies persist due to variations in specular reflection effects (*e.g.*, the distinction between deep blue and light blue). Therefore, we propose a palette-adaptive reconstruction loss, which mitigates the impact of viewpoint-induced color discrepancies:

$$\mathcal{L}_{\text{Palette}} = \sum_{\boldsymbol{r} \in \mathcal{R}} \left\| \hat{\boldsymbol{W}}(\boldsymbol{r}) \cdot \hat{\boldsymbol{P}} - \boldsymbol{C}_{\text{gt}}(\boldsymbol{r}) \right\|_2^2, \tag{11}$$

where $\mathcal{R}$ represents the sampled rays within a training patch, $\boldsymbol{C}_{\text{gt}}$ denotes the RGB ground truth of the segmentated scene, $\hat{\boldsymbol{W}}(\boldsymbol{r})$ is the predicted view-dependent weight for ray $\boldsymbol{r}$, and $\hat{\boldsymbol{P}}$ is the learnable view-invariant palette initialized by $\boldsymbol{P}$. This design assigns the task of learning all corresponding diffuse, highlight, and shadow effects to the weights $\hat{\boldsymbol{W}}(\boldsymbol{r})$. Our palette $\hat{\boldsymbol{P}}$, in addition to providing the base colors, also allows for the input of randomly initialized colors for greater flexibility.

Based on this strategy, the generated weights exhibit multi-view consistency. Therefore, by only modifying the trained palette, we can directly composite locally recolored scenes. Subsequently, the masks obtained during the segmentation stage are utilized to directly replace the corresponding regions in the rendered views, thereby achieving a faithful restoration of the scene. For scenes with multiple classes, a single, shared palette and unified weight maps are used. The distinct color of each object is then controlled by its specific learned weights within the unified weight maps.

### 4.3 Optimization

**Segment Loss.** We compute the multi-class cross-entropy loss between the predicted segmentation map $\hat{\boldsymbol{S}}^l$ and the 2D segmentation map $\boldsymbol{S}^l$ generated by SAM [19] to encourage consistency between the rendered segmentation objects and the provided segmentation map in terms of the primary object. Here, $1 \leq l \leq N_k$ denotes the class index:

$$\mathcal{L}_{\text{Seg}} = - \sum_{\boldsymbol{r} \in \mathcal{R}} \sum_{l=1}^{N_k} \boldsymbol{S}^l(\boldsymbol{r}) \log(\hat{\boldsymbol{S}}^l(\boldsymbol{r})). \tag{12}$$

**Weight Loss.** We calculate the L2 difference between the predicted weights $\hat{\boldsymbol{W}}$ and the 2D sparse weights $\boldsymbol{W}$ to learn the main distribution of the weights, gradually optimizing the prominence of different colors within the scene objects:

$$\mathcal{L}_{\text{Weight}} = \sum_{\boldsymbol{r} \in \mathcal{R}} \left\| \hat{\boldsymbol{W}}(\boldsymbol{r}) - \boldsymbol{W}(\boldsymbol{r}) \right\|_2^2. \tag{13}$$

**Total Loss.** We optimize the overall pipeline by jointly minimizing the two losses mentioned above along with the palette-adaptive reconstruction loss:

$$\mathcal{L}_{\text{Total}} = \lambda_{\text{Seg}} \mathcal{L}_{\text{Seg}} + \lambda_{\text{Edit}} (\lambda_{\text{P}} \mathcal{L}_{\text{Palette}} + \lambda_{\text{W}} \mathcal{L}_{\text{Weight}}), \tag{14}$$

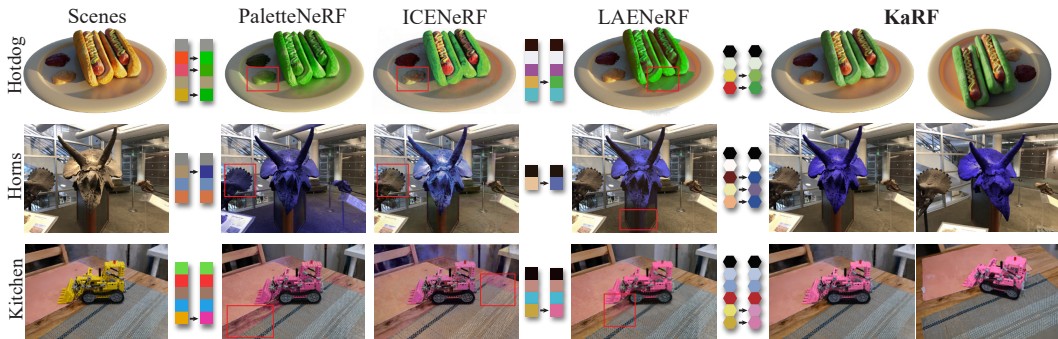

Figure 5: Qualitative comparison with NeRF local color editing methods. The red boxes show the error of the comparison methods on the background.

Table 1: Quantitative comparison with NeRF segmentation methods.

| Scenes | MVSeg [30] | | SA3D [7] | | **KaRF** | |
|---|---|---|---|---|---|---|
| | IoU(%)↑ | Acc(%)↑ | IoU(%)↑ | Acc(%)↑ | IoU(%)↑ | Acc(%)↑ |
| Orchids | 92.7 | 98.8 | 87.9 | 97.8 | **93.2** | **98.9** |
| Leaves | 94.9 | 99.7 | **97.5** | **99.9** | 97.2 | 99.9 |
| Fern | 94.3 | 99.2 | 97.3 | 99.6 | **97.6** | **99.7** |
| Room | 95.6 | 99.4 | 90.4 | 98.6 | **98.3** | **99.8** |
| Horns | 92.8 | 98.7 | 95.4 | 99.2 | **95.4** | **99.2** |
| Fortress | 97.7 | 99.7 | **98.4** | **99.8** | 98.2 | 99.7 |
| Fork | 87.9 | 99.5 | **89.8** | **99.6** | 84.4 | 99.3 |
| Truck | 85.2 | 95.1 | 96.1 | 98.7 | **96.9** | **98.9** |
| Lego | 74.9 | 99.2 | 90.9 | 99.7 | **91.3** | **99.7** |
| Mean | 90.7 | 98.8 | 93.7 | 99.2 | **94.7** | **99.5** |

where $\lambda_{\text{Seg}}$ and $\lambda_{\text{Edit}}$ are set to zero during different stages, while $\lambda_{\text{P}}$ and $\lambda_{\text{W}}$ are hyperparameters for the loss weights.

## 5 Experimental Results

**Datasets.** We test KaRF on three different types of datasets: NeRF-Synthetic [29], a 360° bounded synthetic dataset; Mip-NeRF 360 [1], a 360° unbounded real-world dataset; and LLFF [28], a bounded real-world forward-capture dataset.

**Implementation Details.** In the geometric modeling stage, we use the Adam optimizer with a learning rate of 5e-5, updating based on 2048 rays per iteration, for a total of 120k iterations. For other stages, we use a learning rate of 5e-4 with 1024 rays per update. The segmentation stage involves 2k iterations, while the local recoloring stage involves 5k iterations, during which the learnable palette $P$ is trained only during the final 2k iterations. The hyperparameters $\lambda_{\text{P}}$ and $\lambda_{\text{W}}$ are set to 1 and 1e-1, respectively. All experiments are conducted on a single Nvidia RTX 4090 GPU.

### 5.1 Qualitative Evalution

**Segmentation.** By providing rough reference views of the target object, our proposed KaRF approach can generate fine-grained novel view masks. As illustrated in Figure 4, we compare the proposed KaRF with SA3D [7] and LAENeRF [32] methods in both forward-facing and 360° scenarios, where the second column represents our one or three annotated reference views. The comparison shows that LAENeRF often includes unnecessary objects, as indicated by the red box. Although SA3D shows some advantages in modeling local features, it still struggles with capturing fine texture details of objects, as shown in the yellow box. Additional details are available in the supplementary material.

**Local Recoloring.** Figure 5 presents a comparison of our proposed KaRF method with LAENeRF [32], ICENeRF [23] and PaletteNeRF [22] equipped with the LSeg module [24] across three different datasets. It is worth noting that the results of ICENeRF are taken from its publication [23]. From the experimental results, it is evident that our proposed KaRF method can accurately change the color of the segmented regions without introducing unnecessary color bleeding or artifacts beyond the segmentation boundaries. For example, in the kitchen scene, when recoloring the lego, ICENeRF and PaletteNeRF exhibit noticeable artifacts in the tabletop area. Meanwhile, LAENeRF employs

Table 2: Quantitative comparison with NeRF local color editing methods.

| Datasets | PNF(with LSeg) [22] | ICENeRF [23] | LAENeRF [32] | **KaRF** |
|---|---|---|---|---|
| | MSE↓ | MSE↓ | MSE↓ | MSE↓ |
| LLFF [28] | 0.0056 | 0.0075 | 0.0020 | **0.0002** |
| Blender [29] | 0.0031 | - | 0.0030 | **0.0003** |
| Mip360 [1] | 0.0027 | - | 0.0033 | **0.0002** |
| Mean | 0.0038 | 0.0075 | 0.0028 | **0.0002** |

Table 3: Quantitative comparison of short-range and long-range consistency after local recoloring.

| Consistency | Dataset | PNF(with LSeg) [22] | | LAENeRF [32] | | **KaRF** | |
|---|---|---|---|---|---|---|---|
| | | LPIPS↓ | RMSE↓ | LPIPS↓ | RMSE↓ | LPIPS↓ | RMSE↓ |
| Short-range | LLFF [28] | 0.115 | 0.066 | 0.110 | 0.061 | **0.100** | **0.058** |
| | Blender [29] | 0.206 | 0.239 | 0.209 | 0.238 | **0.201** | **0.234** |
| | Mip360 [1] | 0.228 | 0.103 | 0.228 | 0.095 | **0.218** | **0.090** |
| | Mean | 0.183 | 0.136 | 0.182 | 0.131 | **0.173** | **0.127** |
| Long-range | LLFF [28] | 0.235 | 0.161 | 0.233 | 0.147 | **0.227** | **0.145** |
| | Blender [29] | 0.353 | **0.358** | 0.356 | 0.380 | **0.350** | 0.361 |
| | Mip360 [1] | 0.529 | 0.266 | 0.541 | 0.253 | **0.508** | **0.243** |
| | Mean | 0.372 | 0.262 | 0.377 | 0.260 | **0.362** | **0.250** |

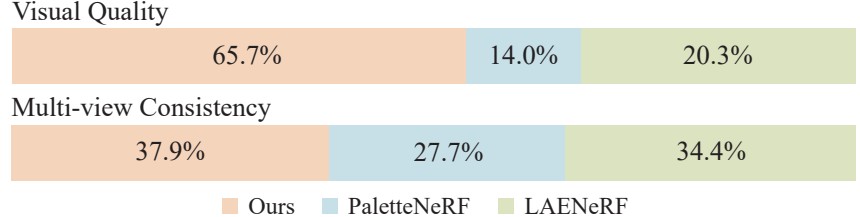

Figure 6: User study results.

a voxel-based segmentation approach, which generates artifacts near the lego, as shown in the red box. Moreover, our proposed KaRF approach excels in maintaining the consistency and plausibility of color adjustments. In the horns scene, our proposed approach better preserves the gloss and tone transitions of the material. In contrast, LAENeRF, due to its limited base colors, modifies similar but distinct colors simultaneously, as it cannot achieve sparse weight decomposition. Other competing methods often result in color deviations after recoloring, especially in reflective areas.

## 5.2 Quantitative Evalution

**Segmentation**. We follow SA3D [7] to use the weakly-supervised dataset provided by SPIn-NeRF [30] to compare our proposed KaRF approach with the MVSeg module in SPIn-NeRF and the SA3D. As shown in Table 1, KaRF achieves the highest average Intersection over Union (IoU) and Accuracy (Acc) across all scenes. This demonstrates the superior segmentation capabilities and robust understanding of scene geometry achieved by our model. For a fair comparison, we follow the SPIn-NeRF benchmark protocol, and the values of SPIn-NeRF and SA3D are taken from [7].

**Local Recoloring**. To measure unintended changes in non-edited regions, similar to LAENeRF [32], we compute the mean squared error (MSE) by comparing these regions in the original image against their state after our recoloring. In Table 2, we present three scenes from three different datasets and compare our proposed KaRF approach with ICENeRF [23], LAENeRF [32] and PaletteNeRF [22] equipped with the LSeg module [24]. It is worth noting that the values of ICENeRF on the LLFF dataset are from its original publication [23], and thus, the foreground mask we use is consistent with that of ICENeRF. Experimental results demonstrate that our approach outperforms other competing methods. This superiority arises from our direct use of precise masks from the segmentation stage to extract palettes and weights for specific local regions, thereby effectively shielding non-segmented areas from being altered during the recoloring process.

To evaluate multi-view consistency, we select pairs of views with intervals of 1 and 7 under short-range and long-range conditions, respectively. Table 3 presents the LPIPS [36] and RMSE results across various scenes after local recoloring with KaRF, PaletteNeRF [22] with the LSeg module [24], and LAENeRF [32]. As can be seen, KaRF achieves the best performance in terms of consistency.

## 5.3 User Study

We conduct a user study comparing our method with PaletteNeRF [22] and LAENeRF [32]. Forty-four participants are presented with pairs of recolored images and videos and asked to make selections

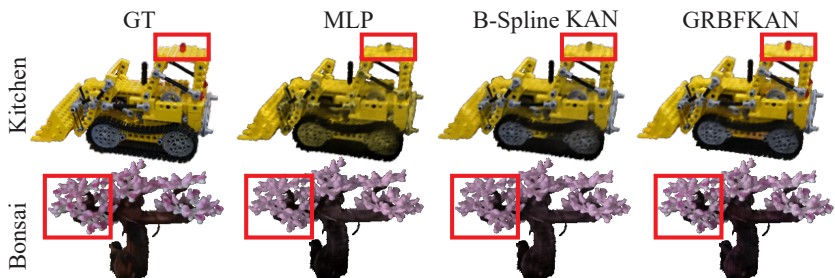

Figure 7: The impact of GRBFKAN. We compare the composite results for the palettes and weights generated by MLP, B-Spline KAN, and GRBFKAN. The red boxes highlight the composition differences.

Table 4: The impact of residual adaptive gating KAN structure.

| Datasets | w/o All | | w/ Gate | | w/o Res | | w/ All | |
|---|---|---|---|---|---|---|---|---|
| | PSNR↑ | SSIM↑ | PSNR↑ | SSIM↑ | PSNR↑ | SSIM↑ | PSNR↑ | SSIM↑ |
| LLFF [28] | 32.77 | 0.961 | 33.24 | 0.964 | 33.50 | 0.966 | **33.52** | **0.966** |
| Blender [29] | 37.54 | 0.960 | 38.30 | 0.980 | 38.68 | 0.984 | **38.95** | **0.988** |
| Mip360 [1] | 30.81 | 0.931 | 30.79 | 0.937 | 31.02 | 0.948 | **31.63** | **0.952** |
| Mean | 33.71 | 0.951 | 34.11 | 0.960 | 34.40 | 0.966 | **34.70** | **0.969** |

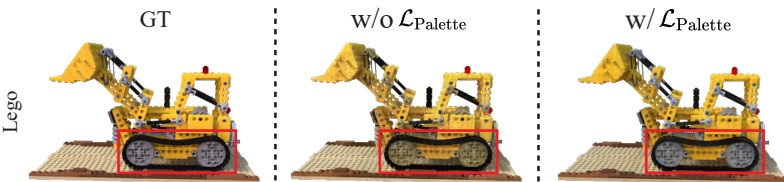

Figure 8: The impact of $\mathcal{L}_{\text{Palette}}$. The red boxes highlight the composition differences.

based on visual quality and view consistency. A total of 528 votes are collected, as shown in Figure 6. The results indicate a clear preference for our method, which is consistently rated as more visually consistent and for its higher quality of local color editing.

### 5.4 Ablation Studies

**Impact of GRBFKAN.** Figure 7 shows GRBFKAN outperforming B-Spline KAN and MLP in palette and weight generation and composition. Unlike MLP, which tends to learn global patterns while ignoring local colors, and B-Spline KAN, which tends to smooth out sharp color details, GRBFKAN learns richer color gradations from identical spatial-level palette and weight inputs, yielding compositions more faithful to the original scenes.

**Impact of Residual Adaptive Gating KAN.** We compare the local recoloring results of stacking only KAN layers (w/o All), introducing gating KAN alone (w/ Gate), introducing gating and the adaptive operator $\mathcal{G}$ without residual connections (w/o Res), and including all components (w/ All), as shown in Table 4. The results demonstrate that our complete structure achieves the best performance in terms of both PSNR and SSIM. Furthermore, the overall structure with residual connections also demonstrates significant improvement in terms of convergence speed and loss.

**Impact of $\mathcal{L}_{\text{Palette}}$.** It can be observed in Figure 8 that the model without the palette-adaptive reconstruction loss (w/o $\mathcal{L}_{\text{Palette}}$) exhibits a noticeable deviation in the color reproduction of local regions when compared to the original scene.

## 6 Conclusion

We have proposed KaRF, a novel weakly-supervised NeRF local color editing method that leveraged the KAN. By designing a unified two-stage framework, our proposed KaRF approach effectively achieved both scene segmentation and realistic editing functionalities. Innovatively, we have introduced KAN-based task-specific radiance fields and designed the residual adaptive gating KAN structure, which significantly enhances the computational efficiency and accuracy of the model. Extensive experimental results have demonstrated that the proposed KaRF algorithm outperforms many state-of-the-art methods for segmentation and local recoloring tasks in terms of both visual performance and quantitative metrics.

## Acknowledgments and Disclosure of Funding

This research is supported in part by National Natural Science Foundation of China under Grant 62471199, 62020106011, and 62271414, in part by National Foreign Experts Program under Grant S20250222, in part by National Natural Science Fund for Excellent Young Scientists Fund Program (Overseas), and in part by National Key R&D Program Project under Grant 2023YFC3806003.

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
