# OpenReview forum: "KaRF: Weakly-Supervised Kolmogorov-Arnold Networks-based Radiance Fields for Local Color Editing"
_NeurIPS.cc/2025/Conference — NeurIPS 2025 poster_

### Official Review · Reviewer_JVR7 · 2025-06-13

**Clarity:** 2
**Significance:** 2
**Originality:** 2
**Rating:** 3
**Confidence:** 4

**Summary:**

This paper addresses the task of color editing in neural radiance fields (NeRFs), with a specific focus on combining local editing and palette-based editing. Traditional palette-based approaches typically lack spatial precision unless paired with segmentation masks. The authors overcome this limitation by using SAM-generated masks to supervise the learning of a local 3D segmentation field, which is then used to perform palette-based decomposition in localized regions.

Starting from a pre-trained NeRF, the method leverages its multi-channel features to train a per-point segmentation based on KAN (upgraded to GRBFKAN), followed by palette decomposition on the segmented regions. The key contribution lies in enabling localized palette decomposition, which allows for richer and more controllable local edits.

The authors demonstrate improved local editing capabilities compared to a basic KAN-based baseline.

**Questions:**

How many SAM masks are actually needed for training?

How is SAM being prompted?

Would it be possible to visualize or quantify the separation between what is modeled as view-independent (via W) and view-dependent (residuals)?

Clarification is needed on the input-output relationship between the segmentation map and the palette weights W — specifically, how (and whether) view direction influences both.

It is unclear whether the parts of the scene not belonging to the segmented object (e.g., background or other classes) are also palette-decomposed. If they are not palette-decomposed, what does KaRF learn for those regions? Just a single base color per pixel/point?
Or does KaRF learn separate palette decompositions for each segmentation class (e.g., one palette for object 1, another for object 0)?


I would be open to accepting this paper if the following conditions are addressed:

-The authors demonstrate that the segmentation module can operate effectively with only a single input mask or single user interaction, and provide both quantitative and qualitative evidence of its generalization capabilities.

-There is clear empirical justification—both qualitative and quantitative—for using KAN or GRBF-KAN over standard MLPs. It should be demonstrated that the improved performance and enhanced local editability are due to the KAN architecture itself, and not simply the result of increased supervision(SAM) compared to other methods.

-The paper includes more comprehensive 360° visual evidence, particularly on the Mip-NeRF 360 dataset, showing that the method maintains multi-view consistency and handles well view-dependent effects.

**Ethical Concerns:**

["NO or VERY MINOR ethics concerns only"]

**Final Justification:**

I am raising my score to a 3, since they showed evidence of the KAN being more useful than the MLPs.
Still, I don't like how the ablation studies were done, throughout all the datasets, choosing only a few scenes per dataset, sometimes only one.

Also, since nothing is done about view-dependent effects but they claim it is learned through the black-and-white palette, I would have liked videos on the Mip360 datasets (the most complex) showcasing how the recoloring is working even in 360° scenarios.

**Limitations:**

Yes

**Quality:**

1

**Strengths And Weaknesses:**

Supervision Level and Use of SAM Masks:
The method relies on segmentation masks generated by SAM on multiple views to train a 3D segmentation field via a KAN-based radiance field. While the authors frame this as “weak supervision,” the requirement of 2D masks from multiple viewpoints constitutes a considerably stronger form of supervision than prior methods. ICE-NeRF uses sparse scribbles on a single image and IReNe relies only on a single segmentation mask. In contrast, KaRF appears to need multiple SAM masks to function effectively. Additionally, the paper does not clearly state how many views are used — Figure 4 shows two, but it's unclear if that's just for illustration. Calling this “weak supervision” is, in my view, somewhat misleading.

Lack of Justification for Using KANs:
The paper introduces using KANs, specifically GRBF-KANs, as an architectural innovation for both segmentation and recoloring. While ablations show that their residual adaptive gating improves over basic KANs and B-spline variants, there is no direct numerical comparison with standard MLPs, despite MLPs being used in all previous NeRF-based editing frameworks. Since the architectural shift to KANs is the major contribution, this omission weakens the justification for their use.

Viewdep Inconsistency:
The paper states that segmentation is view-invariant (Eq. 7), but Figure 3 shows the view direction as an input to the segmentation module, creating an inconsistency. This should be clarified, as it raises doubts about whether the segmentation field truly disentangles geometry from appearance.

More importantly, I’m confused by the treatment of view direction in the palette weight prediction. In the text, the palette weights W are defined as a function of view direction (Eq. 10). That’s acceptable, although in previous SOTA methods, like PaletteNeRF, the palette weights were view-invariant. In other papers, like IReNe, there was a focus on explicitly dividing specular and diffuse components. Now, if there’s a need to make the weights not be affected too much by view-dependent color, I would have assumed Equation 11 to be used with view-invariant palette weights W — to say: "learn as much of the color as you can through W, so the view direction can then learn only the residual." But that doesn’t make sense if W is not direction-invariant.
If W is view-dependent, then Equation 11 is effectively a standard RGB reconstruction loss — so why not call it that? I believe a proper RGB reconstruction loss is actually missing in KaRF (if Equation 11 is not intended for that) to fully reconstruct the scene. I would have expected both a photometric RGB loss and a loss that encourages W to learn as much of the diffuse color as possible.
As written, Equation 11 seems like a standard RGB reconstruction loss, and Equation 13 is just to push W to learn the main color components. If that’s the case, I wouldn’t call Equation 11 "L_palette" — it’s more accurate to call it "PhotometricLoss".
I would like the authors to clarify this point.

Limited Scope of Ablation Studies:
Most ablation studies are conducted on single scenes only — e.g., LPalette is tested only on Lego, GRBF-KAN only on Kitchen, and time comparison only on Fern. This narrow scope makes it difficult to assess the generalizability of the architectural improvements or the claimed efficiency. Broader evaluation would strengthen the evidence for the proposed choices.

Limited Qualitative Evaluation on Challenging Scenes:
I would have appreciated more qualitative results, especially full 360° visualizations on the Mip-NeRF 360 dataset. Since reflections and complex lighting effects are critical challenges in recoloring, it is important to see how well the method performs in those settings. Static frame comparisons are insufficient to assess view consistency in reflective surfaces. I find it suspicious that no complete 360° video of Mip-NeRF 360, or even images showing significant viewpoint changes, were included either in the main paper or the supplementary material. This omission makes me think the method may not generalize well in 360° cases.

---

> ### Author Rebuttal · Authors · 2025-07-30
>
> Thank you for your thoughtful comments and constructive feedback. We conduct extensive experiments to address your feedback (including excellent results with 1/3 input masks, more comprehensive ablation studies, and 360° videos of mip360). However, due to the official rule against adding images in the rebuttal, we provide ample quantitative results and describe some of the visual outcomes. We promise that if the paper is accepted, we will include more visual experiments.
>
> **Q1**: Supervision Level and Use of SAM? How many masks for training? How to prompt SAM?
>
> **Response**: We define weak supervision by two key properties: the use of sparse and imperfect labels. Our method requires only 1-3 coarse masks per scene (one for forward-facing like LLFF, three for 360° scenes like Mip360). These masks are automatically generated by prompting SAM with points and are used directly without any manual refinement.
>
> Though we may use more than one mask, our supervision is more efficient than prior work. Unlike IReNe, which requires precise manual maps and new reference images for multiple edits, KaRF uses coarse, auto-generated masks and modifies a single learned palette. Furthermore, unlike ICENeRF (often requires at least two masks: foreground and background), whose foreground mask requirement scales with the number of colors being edited, KaRF edits multiple objects within the same sparse set of masks.
>
> We apologize for the confusion caused by Figure 4. The dense-mask setup shown was used exclusively to adhere to the SPIn-NeRF benchmark protocol for a fair comparison and does not represent our method's standard workflow.
>
> **Q2**: Viewdep Inconsistency? the input-output relationship?
>
> **Response**: We apologize for the lack of clarity in Figure 3. The confusion stems from the legend: the view direction input is used only for local recoloring, not segmentation. Its enclosure in the red dashed box visually signifies its exclusive role in the recoloring process.
>
> To directly clarify the input-output relationships for each stage:
>
> 1. Segmentation Stage (View-Invariant): Inputs: 3D position + prior features. Output: consistent segmentation maps. This stage learns the intrinsic geometry and semantic class of the scene, which is independent of the viewing angle.
>
> 2. Local Recoloring Stage (View-Dependent): Inputs: 3D position + view direction + prior features. Output: consistent weight maps and a palette. Composites colors by location and viewing angle to reconstruct view-dependent effects (gloss, shading, reflections).
>
> **Q3**: Clarification on the L_palette and view-dependent/independent effects?
>
> **Response**: Unlike PaletteNeRF/IReNe, KaRF's view-dependent weights W model all color effects (diffuse, specular) with a view-invariant palette P. This is enabled by LoCoPalette (including black/white bases), supporting our two-phase training:
>
> 1. First 2k Iterations: Lweight provides a coarse, view-invariant guide to color distribution. Simultaneously, Lpalette (with a fixed palette P) acts as a strong view-dependent loss, enabling W to learn all necessary details for ground truth approximation. This includes utilizing white base for specular highlights and black base for shadows, with all diffuse, highlight, and shadow weights learned within W.
>
> 2. Last 1k Iterations: Palette P becomes learnable. The L_palette now also adaptively fine-tunes the base colors in P to achieve the best possible color fidelity for the reconstructed image.
>
> You have made an excellent point regarding the naming of L_palette. To better reflect its dual role, we will rename it to Palette-Adaptive RGB Reconstruction Loss.
>
> To visualize and quantify effect separation, our visualizations confirm that diffuse weights stay stable when the viewing angle changes for a fixed 3D point. Conversely, white and black weights vary significantly with view direction, precisely how view-dependent highlights and shadows are formed. An ablation study in Tab 1, with image contrast amplified 4x to highlight specular effects, quantifies this impact, demonstrating significant improvement from view-dependent weights in capturing complex lighting compared to a view-invariant model.
>
> **Tab 1**: Weight W.
> |Type|PSNR↑|SSIM↑|
> |-|-|-|
> |View-Invariant|25.69|0.961|
> |View-Dependent|**27.42**|**0.972**|
>
> **Q4**: Strategy for background or other classes?
>
> **Response**: KaRF is a modular framework that exclusively edits user-specified foreground objects, leaving the background unmodified. During training, background regions learn to select only a black palette base, effectively isolating them from the recoloring process.
>
> For scenes with multiple classes, a single, shared palette and a unified weight field W are used. The distinct color of each object is then controlled by its specific learned weights within this unified weight field.
>
> **Q5-**: Single input mask evidence?
>
> **Response**: For LLFF, we require only a single mask to achieve high-quality segmentation. For Blender and Mip360, we use three masks from minimally-overlapping viewpoints.
>
> We performed extensive experiments to validate this capability. The results of Tab 2 show that the metrics achieved with this sparse-mask supervision are nearly identical to the results from using a much larger set of masks. Qualitatively, the generated segmentations and the final local edits are visually indistinguishable from the results obtained with denser supervision. This demonstrates our model's strong generalization ability, allowing it to learn a complete and accurate segmentation field from extremely sparse signals.
>
> **Tab 2**: Sparse masks input (IoU↑ Acc↑).
> |Scene|1/3 masks|all masks|
> |-|-|-|
> |LLFF-Horns|0.951 0.992|0.954 0.992|
> |LLFF-Flower|0.960 0.991|0.963 0.991|
> |Blender-Chair|0.970 0.997|0.982 0.998|
> |Blender-Hotdog|0.962 0.996|0.964 0.996|
> |Mip360-Bonsai|0.868 0.994|0.879 0.994|
> |Mip360-Kitchen|0.961 0.994|0.968 0.995|
>
> **Q6-**: Impact of GRBFKAN, L_palette, residual adaptive gating KAN and time comparison?
>
> **Response**: We sincerely apologize that our initial experiments were not sufficient to fully demonstrate the generality and superiority of our design choices.
>
> 1. Impact of GRBFKAN: To demonstrate that our performance gains stem from the KAN architecture itself and not simply the supervision level, we conducted a broad comparison of MLP, B-Spline KAN, and GRBFKAN across six scenes from three datasets. The results of Tab 3 show that GRBFKAN consistently yields the best PSNR and SSIM, confirming its superior ability to model high-frequency color details compared to the smoother B-splines and global MLP. The results of Tab 4 confirm that GRBFKAN achieves the best average IoU and Acc, demonstrating its enhanced ability to learn complex geometric and semantic boundaries. Qualitatively, GRBF KAN also produces sharper segmentation boundaries and more accurate color compositions that are faithful to the original scenes.
>
> **Tab 3**: Impact of the GRBFKAN (PSNR↑ SSIM↑).
> |Scene|MLP|BspineKAN|GRBFKAN|
> |-|-|-|-|
> |Horns|35.14 0.971|35.66 0.968|**35.97 0.977**|
> |Flower|32.13 0.964|32.94 0.966|**33.52 0.966**|
> |Chair|33.76 0.986|35.07 0.984|**35.12 0.985**|
> |Hotdog|36.49 0.967|36.85 0.972|**38.95 0.988**|
> |Bonsai|36.96 0.985|36.97 0.977|**38.31 0.989**|
> |Kitchen|27.65 0.934|30.29 0.945|**31.63 0.952**|
> |Mean|33.69 0.968|34.63 0.969|**35.58 0.976**|
>
> **Tab 4**: Impact of the GRBFKAN (IoU↑ Acc↑).
> |Scene|MLP|BspineKAN|GRBFKAN|
> |-|-|-|-|
> |Horns|0.949 0.992|0.946 0.991|**0.951 0.992**|
> |Flower|**0.961 0.991**|0.960 0.991|0.960 0.991|
> |Chair|0.967 0.997|0.967 0.997|**0.970 0.997**|
> |Hotdog|0.949 0.994|0.946 0.994|**0.962 0.996**|
> |Bonsai|0.862 0.994|0.849 0.993|**0.868 0.994**|
> |Kitchen|0.955 0.994|0.952 0.993|**0.961 0.994**|
> |Mean|0.941 0.994|0.937 0.993|**0.945 0.994**|
>
> 2. Impact of L_palette: The results from three new scenes confirm that including our L_palette loss consistently and significantly improves reconstruction quality (PSNR/SSIM) and prevents color deviation across all dataset types:
>
> **Tab 4**: Impact of L_palette (PSNR↑ SSIM↑)
> |Scene|w/o L|w/ L|
> |-|-|-|
> |Flower|25.57 0.933|**33.52 0.966**|
> |Lego|28.95 0.940|**31.72 0.958**|
> |Kitchen|30.30 0.888|**31.63 0.952**|
>
> 3. Extended Time Comparison: We provide runtimes for three scenes from different datasets, demonstrating the stability and generalizability of our method's efficiency:
>
> **Tab 5**: Processing Times
> |Dataset|Time|
> |-|-|
> |LLFF|9.93min|
> |Blender|9.97min|
> |Mip360|10.14min|
> |Mean|10.01min|
>
> 4. Impact of Residual Adaptive Gating KAN: We performed new ablations on three scenes for both segmentation and recoloring. The results of Tab 6 and 7 show our full design is consistently optimal. While its segmentation score is similar to w/o Res, it is significantly better than other variants, and as shown in Supp. Fig. 7, it converges much faster.
>
> **Tab 6**: Impact of structure (IoU↑ Acc↑)
> |Scene|w/o All|w/ Gate|w/o Res|w/ All|
> |-|-|-|-|-|
> |Leaves|0.144 0.948|0.000 0.939|0.969 0.998|**0.970 0.998**|
> |Bonsai|0.090 0.966|0.015 0.963|0.868 0.994|**0.868 0.994**|
> |Lego|0.037 0.973|0.000 0.972|0.888 0.997|**0.891 0.997**|
>
> **Tab 7**: Impact of structure (PSNR↑ SSIM↑)
> |Scene|w/o All|w/ Gate|w/o Res|w/ All|
> |-|-|-|-|-|
> |Flower|32.77 0.961|33.24 0.964|33.50 0.966|**33.52 0.966**|
> |Kitchen|30.81 0.931|30.79 0.937|31.02 0.948|**31.63 0.952**|
> |Hotdog|37.54 0.960|38.30 0.980|38.68 0.984|**38.95 0.988**|
>
> **Q7-**: Mip360 evidence?
>
> **Response**: We apologize for the oversight in our initial submission's video presentation. Our method's superior multi-view consistency is already demonstrated quantitatively in Table 3 of the supplementary material, and we have now rendered a full 360° video for the Mip-360 dataset as further visual proof, which we are unable to upload here.
>
> **Q8**: Ethical considerations?
>
> **Response**: First, we use only public, non-sensitive benchmark datasets. Second, our supplementary material (Sec. 10) discusses both the positive societal impacts and the potential for misuse.

---

> > ### Comment · Reviewer_JVR7 · 2025-08-02
> >
> > Clarification on Q3 response. You mean "a VIEW-VARIANT palette" right? If it's view-invariante how can it learn specular component?

---

> > > ### Author Response · Authors · 2025-08-02
> > >
> > > Thank you very much for the follow-up question. We provide a clearer explanation on this question 3:
> > >
> > > Your understanding is spot-on. Indeed, for the model to learn view-dependent effects such as specular reflections, some component within it must be able to perceive and respond to changes in viewing direction. In our KaRF framework, this view-dependency is learned entirely by the weight set W, while the palette P itself remains view-invariant (it can be thought of as a collection of base colors).
> > >
> > > The specifics are as follows:
> > >
> > > 1. **View-Dependent Weight**: For any given 3D point, the weight W produced by the network changes dynamically according to the viewing direction.
> > >
> > > 2. **View-Independent Palette**: The palette is a set of colors that is shared across the entire scene and does not change with the viewing angle. It typically contains the object's diffuse colors, as well as pure white and black base colors to represent lighting and shadows. Although the palette is adaptively adjusted during the final 1k training iterations, the optimization at this stage is still for a view-independent palette.
> > >
> > > 3. **Composition Method**: The final pixel color is synthesized through the weighted sum of the view-dependent weights and the view-independent palette.
> > >
> > > 4. **For instance**: when observing a glossy surface from a specular reflection angle, our network learns to assign a very high weight to the white base color within the palette, thereby producing a highlight effect. Conversely, when the same point is viewed from other angles, the weight for this white base color decreases significantly, allowing the weights for the object's diffuse colors to become dominant. It is this dynamic change in the weights, rather than a change in the palette, that enables the realistic modeling of view-dependent effects.

---

> ### Comment · Reviewer_JVR7 · 2025-08-03
>
> Ok thank you now it's clear, it was my bad, I was thinking for the palette and the weights as one single thing, but they are separed, and the W is the one view-dep.
>
> Is there any reason why not all the scenes of mip360 were taken a subjects for the experiments?
>
> I would have valued more experiments only done on mip360, than this division of some LLFF, some Synthetic, and some mip360.
> For example the garden scene, with the table, would have been a nice test to check the consistency of the view-dependent effects learned on the table

---

> > ### Author Response · Authors · 2025-08-05
> >
> > Thank you for your question. We apologize for the delayed response, as we took the time to supplement the experiments on all Mip360 scenes that you mentioned. We would like to provide the following explanation regarding our choice of datasets and the selection of scenes from the Mip360 dataset:
> >
> > 1. **Dataset Selection**: Our approach was tested on three different types of datasets: NeRF-Synthetic, LLFF, and Mip-NeRF 360. This cross-dataset evaluation strategy is intended to demonstrate the versatility and robustness of KaRF across a variety of scenarios, covering bounded synthetic scenes, bounded forward-facing real-world scenes, and unbounded 360° real-world scenes. The reason for selecting these three datasets is that we directly follow recent works, such as LAENeRF and ICENeRF, and adopt their evaluation method of reporting results on these diverse benchmarks. To ensure a fair comparison, we also used the same scenes they did.
> >
> > 2. **Scene Selection**: For the Mip360 dataset, we specifically selected the Bonsai, Kitchen, and Room scenes for our experiments. This choice was not because our method is incapable of handling other scenes like Garden, but rather to maintain consistency with the evaluation protocols of the methods we are comparing against (e.g., LAENeRF and ICENeRF). This allows for a strictly controlled quantitative and qualitative comparison.
> >
> > 3. **Additional Mip360 Results**: For a more comprehensive evaluation, we are providing the results for all additional scenes from the Mip360 dataset here. This includes the Garden (segmentation object: the table), Counter (segmentation object: the blue glove), Stump (segmentation object: the stump), and Bicycle (segmentation object: the Bicycle) scenes. The results of Tab 1 indicate that the foreground scene composited from the palette and weights learned by KaRF achieves high PSNR and SSIM values when compared to the ground-truth foreground scene. This further demonstrates that our method can generate high-fidelity, locally recolored scenes while also ensuring the consistency of view-dependent effects. The results in Tab 2 show that KaRF is capable of producing high-quality scene segmentation masks in these scenes. If the paper is accepted, we will incorporate the qualitative and quantitative results for all scenes from the Mip360 dataset into the main paper.
> >
> > **Tab 1**: Composite quality using palette and weights for all additional Mip360 dataset scenes.
> > |Scene|PSNR↑ SSIM↑|
> > |-|-|
> > |Garden|33.92 0.976|
> > |Counter|39.89 0.990|
> > |Stump|27.31 0.915|
> > |Bicycle|28.23 0.920|
> >
> > **Tab 2**: Segmentation quality for all additional scenes in the Mip360 dataset.
> > |Scene|IoU↑ Acc↑|
> > |-|-|
> > |Garden|0.986 0.999|
> > |Counter|0.985 0.999|
> > |Stump|0.912 0.983|
> > |Bicycle|0.908 0.975|

---

### Official Review · Reviewer_1rRd · 2025-06-25

**Clarity:** 3
**Significance:** 2
**Originality:** 2
**Rating:** 4
**Confidence:** 3

**Summary:**

This paper introduces an alternative backbone for NeRF by replacing the MLP with a KAN-based network for local color editing. Given the flexible activation prediction of KAN the authors aim to improve segmentation with finer edges and more precise color editing than MLP based NeRF. The method however added architectural complexity significantly, and the authors did not specify the runtime of the model but simply state certain design to faciliate the computational overhead. Additionally, the method lacks rigorous analysis of scalability and training stability beyond the shown dataset examples.

**Questions:**

See weakness.

**Ethical Concerns:**

["NO or VERY MINOR ethics concerns only"]

**Final Justification:**

I acknowledge the effort put in by the authors to address my concerns. I update my rating to borderline accept.

**Limitations:**

yes

**Quality:**

2

**Strengths And Weaknesses:**

The given implementation details are not sufficient to reproduce the method without the release of the codebase. The author also did not specify whether they plan to release the code.
The authors did not specify how many datasets were used for experiments, but merely explained the number of types of scenes. The authors also did not specify the change in implementation for different types of NeRF scenes.
The runtime comparison is not extensive and significant enough through simply providing "roughly 10 min" without specifying the type of scenes.
The use of limited cross-dimensional interaction by KAN can create interdependencies between view directions and 3d locations compared to fully connected MLPs. How does the model handle the view-dependent effects such as specularity or transparency in editing? How would the model behave for scenes with complex lighting?
Overall, the reason to adapt KAN for the specific task of localized NeRF recoloring is not well conveyed through the writing of this paper, and the authors have not shown enough results to demonstrate superiority of the method.

---

> ### Author Rebuttal · Authors · 2025-07-30
>
> Thank you for the questions you have raised. We will address each of them in turn below.
>
> **Q1**: The given implementation details are not sufficient to reproduce the method without the release of the codebase. The author also did not specify whether they plan to release the code.
>
> **Response**: We sincerely apologize that the details in our paper and supplementary material were not sufficient to fully address your concerns about reproducibility.
>
> We endeavored to provide the key information required to reproduce our method. To be as clear as possible, we have detailed the following in the Implementation Details subsection of section 5: The optimizer used for all stages. The learning rates and batch sizes for the different training stages. The specific number of iterations for each training stage. The weight hyperparameters for all loss functions. Furthermore, in section 4, we provided a detailed network architecture diagram (Figure 3), which includes our core residual adaptive gating KAN structure. We also supplied the precise mathematical definitions for all loss functions and the equations for segmentation and local recoloring. We believe this information provides a solid foundation for reimplementation.
>
> Most importantly, we want to state our open-source commitment unequivocally. As mentioned in our response to the NeurIPS checklist 5, we will make our work fully accessible. Upon acceptance of the paper, we will immediately release the complete source code, all pre-trained models, and the pre-processed data to ensure full reproducibility.
>
> **Q2**: The authors did not specify how many datasets were used for experiments and the change in implementation for different types of NeRF scenes.
>
> **Response**: We apologize for any lack of clarity in our paper. Regarding the datasets used, we would like to clarify that we specified the three distinct datasets in the datasets subsection of section 5. The datasets are: NeRF-Synthetic (Blender), Mip-NeRF 360, and LLFF. For our qualitative evaluation (Figure 5), we selected one representative scene from each of these datasets. Similarly, our quantitative evaluation (Table 2) was conducted on a total of nine scenes drawn from across these same three datasets to ensure a comprehensive assessment.
>
> Regarding your second point on implementation changes for different NeRF scene types, we thank you for raising this. We provided a single, unified set of implementation details because our proposed KaRF framework demonstrates excellent robustness and generalization. The same set of hyperparameters and training configurations was applied across all three distinct dataset types without needing any scene-specific or dataset-specific tuning. For the NeRF-Synthetic, LLFF, and Mip-NeRF 360 datasets, the learning rates, iteration counts, and loss weights listed in the implementation details subsection were used consistently.
>
> **Q3**: The runtime comparison is not extensive and significant enough through simply providing "roughly 10 min" without specifying the type of scenes.
>
> **Response**: We apologize if our description of the runtime was not sufficiently detailed. We chose the Fern scene for our exemplary comparison, a practice aligned with other methods such as LAENeRF, to provide a fair time comparison on a common and publicly available benchmark.
>
> More importantly, we wish to clarify that the runtime for our method is highly stable and consistent across different scenes and dataset types. This is because the total processing time is primarily determined by the fixed number of training iterations and the number of rays processed per batch, rather than the visual complexity of the content of the scene. In our experimental setup, the training iterations and other settings are kept constant for all datasets, including NeRF-Synthetic, LLFF, and Mip-NeRF 360. As a result, the time required for local color editing remains nearly identical regardless of the scene. To provide concrete evidence, Tab 1 shows the processing times for several representative scenes from different datasets:
>
> **Tab 1**: Processing Times.
> |Dataset|Time|
> |-|-|
> |LLFF|9.93min|
> |Blender|9.97min|
> |Mip360|10.14min|
> |Mean|10.01min|
>
> **Q4**: How does the model handle the view-dependent effects such as specularity or transparency in editing? How would the model behave for scenes with complex lighting?
>
> **Response**: While individual edges in a KAN learn univariate functions, we would like to clarify that this does not preclude the network as a whole from learning complex, cross-dimensional dependencies, such as those between 3D locations and view directions. Our model effectively handles view-dependent effects through the following mechanisms:
>
> 1. Explicit View Direction Input: In the local recoloring stage, which is responsible for effects like specularity and transparency, we provide the view direction as an explicit input to the network, following the successful practice of the original NeRF. This is formally shown in Equation (10) of our paper.
>
> 2. Expressive Power of the Composed Network: Although each edge activation is a univariate function, the full network, through its multi-layer composition, is fully capable of learning complex mappings from high-dimensional inputs to high-dimensional outputs. Our proposed residual adaptive gating KAN structure is designed to learn these non-linear dependencies between position and view direction effectively.
>
> 3. Qualitative Evidence: Our experimental results demonstrate the effectiveness of this design. In the Horns scene in Figure 5 of the main paper, our method successfully preserves the surface gloss and tonal transitions after changing the color of the horns. This gloss is a classic view-dependent effect, and its preservation proves that our model accurately learned and reproduced this dependency, a task where other methods like LAENeRF show visible artifacts.
>
> Regarding the behavior of our model in scenes with complex lighting: The Mip-NeRF 360 dataset is representative of the scenes you describe, containing unbounded, real-world environments with complex lighting, shadows, and reflections. Our method achieves excellent editing results across multiple scenes from this challenging dataset, such as the Kitchen scene (main paper Figure 5) and the Bonsai scene (supplementary material Figure 11). In these scenarios, KaRF not only accurately replaces the color of the target object but also naturally integrates the new color into the existing lighting environment of the scene, correctly rendering the resulting highlights, shading, and shadows on the surface of the object.
>
> **Q5**: The reason to adapt KAN for the specific task of localized NeRF recoloring is not well conveyed through the writing of this paper, and the authors have not shown enough results to demonstrate superiority of the method.
>
> **Response**: We sincerely apologize that our initial manuscript did not sufficiently convey the core motivations for choosing the KAN architecture. We appreciate you raising these critical points, and we hope to offer a much clearer justification here. Our primary reasons for adapting KAN for the specific task of localized NeRF recoloring are as follows:
>
> 1. Superior Non-linear Expressiveness: Local color editing is an inherently complex function-fitting problem. KANs possess stronger function approximation capabilities and greater expressive power than traditional MLPs, making them better suited for this task.
>
> 2. Precise Modeling of High-Frequency Details: We further modified the KAN architecture to use GRBF instead of B-splines. This decision was driven by the fact that local editing often involves high-frequency details like specular highlights and sharp edges. Compared to the smoothing properties of B-splines, GRBFs can more flexibly adapt to and reconstruct these rapidly changing local features.
>
> 3. Higher Training Efficiency and Stability: Figure 7 in our supplementary material provides empirical evidence for the advantages of GRBF KAN. In both the segmentation and local recoloring stages, the training loss for GRBF KAN is consistently lower and converges faster than that of both MLP and B-Spline KAN architectures.
>
> To address your valid concern about the lack of sufficient results, we have conducted a broad set of new quantitative experiments to systematically demonstrate the superiority of our GRBF KAN architecture. We selected two representative scenes from each of our three datasets and performed a comprehensive comparison between GRBF KAN, B-Spline KAN, and MLP.
>
> 1. Local Recoloring Task: We evaluate the reconstruction quality after palette synthesis. The results in Tab 2 show that GRBFKAN outperforms B-Spline KAN and MLP across all tested scenes in terms of PSNR and SSIM.
>
> **Tab 2**: Impact of the GRBFKAN (PSNR↑ SSIM↑).
> |Scene|MLP|BspineKAN|GRBFKAN|
> |-|-|-|-|
> |LLFF-Horns|35.14 0.971|35.66 0.968|**35.97 0.977**|
> |LLFF-Flower|32.13 0.964|32.94 0.966|**33.52 0.966**|
> |Blender-Chair|33.76 0.986|35.07 0.984|**35.12 0.985**|
> |Blender-Hotdog|36.49 0.967|36.85 0.972|**38.95 0.988**|
> |Mip360-Bonsai|36.96 0.985|36.97 0.977|**38.31 0.989**|
> |Mip360-Kitchen|27.65 0.934|30.29 0.945|**31.63 0.952**|
> |Mean|33.69 0.968|34.63 0.969|**35.58 0.976**|
>
> 2. Segmentation Task: As shown in Tab 3, GRBFKAN consistently outperforms both B-Spline KAN and MLP in average IoU and Acc. This provides strong empirical support for our claim that GRBFKAN possesses a superior capacity for learning scene geometry and precise semantic boundaries.
>
> **Tab 3**: Impact of the GRBFKAN (IoU↑ Acc↑).
> |Scene|MLP|BspineKAN|GRBFKAN|
> |-|-|-|-|
> |LLFF-Horns|0.949 0.992|0.946 0.991|**0.951 0.992**|
> |LLFF-Flower|**0.961 0.991**|0.960 0.991|0.960 0.991|
> |Blender-Chair|0.967 0.997|0.967 0.997|**0.970 0.997**|
> |Blender-Hotdog|0.949 0.994|0.946 0.994|**0.962 0.996**|
> |Mip360-Bonsai|0.862 0.994|0.849 0.993|**0.868 0.994**|
> |Mip360-Kitchen|0.955 0.994|0.952 0.993|**0.961 0.994**|
> |Mean|0.941 0.994|0.937 0.993|**0.945 0.994**|

---

> > ### Comment · Reviewer_1rRd · 2025-08-05
> >
> > Thank you for addressing the listed concerns in detail! I appreciate the effort made by the authors to provide the additional results. Regarding the additionally provided results, which metric is used to evaluate the multi-view consistency regarding the recolored palette? Note that recoloring an existing scene requires consistency in the edited results.

---

> > > ### Author Response · Authors · 2025-08-05
> > >
> > > Thank you very much for your follow-up questions and for the detailed review of our paper! We would like to clarify that in our previous supplementary experiments, we primarily provided metrics related to the quality of palette and weight composition and segmentation accuracy for the local recoloring task within our ablation studies. Regarding your question about "which metric is used to evaluate the multi-view consistency of the recolored palette," we presented the results in Table 3 of the submitted supplementary material. For your convenience, we have included the relevant content from that table below as Tab 1 and provide the following detailed explanation:
> > >
> > > 1. **Metric Description**: We fully agree with your point that successful scene editing must guarantee a high degree of consistency in the edited results across multi-views. To evaluate this rigorously and quantitatively, we have provided a comprehensive quantitative comparison in Tab 1. Specifically, we adopted two widely recognized and authoritative metrics to measure multi-view consistency: LPIPS and RMSE.
> > >
> > > 2. **Evaluation Conditions**: For a comprehensive evaluation, we considered both short-range and long-range changes in viewpoint. Specifically, we selected pairs of views with intervals of 1 and 7, respectively, for comparison.
> > >
> > > 3. **Evaluation Methods**: We compared KaRF with PaletteNeRF (with LSeg) and LAENeRF across multiple scenes from three different types of datasets (i.e., LLFF, Blender, and Mip360).
> > >
> > > 4. **Result Analysis**: As shown in Tab 1, our KaRF method achieved the best average performance on both the LPIPS and RMSE metrics under both short-range and long-range conditions. This conclusively demonstrates that after performing local color editing on a scene, our method is able to maintain excellent visual consistency across different views.
> > >
> > > We hope the explanation above fully answers your questions regarding the evaluation of multi-view consistency. Thank you again for your valuable time and constructive feedback!
> > >
> > > **Tab 1**: Per-scene LPIPS and RMSE for local recoloring using KaRF, PaletteNeRF with LSeg, and LAENeRF.
> > > | **Consistency** | **Dataset** | **Scene** | **PNF(with LSeg) LPIPS↓** | **PNF(with LSeg) RMSE↓** | **LAENeRF  LPIPS↓** | **LAENeRF  RMSE↓** | **KaRF LPIPS↓** | **KaRF RMSE↓** |
> > > | :--- | :--- | :--- | :--- | :--- | :--- | :--- | :--- | :--- |
> > > | **Short-range** | **LLFF** | Flower | 0.088 | 0.034 | 0.083 |0.024 | **0.076** | **0.021** |
> > > | | | Horns | 0.147 | 0.077 | 0.141 | 0.070 | **0.131** | **0.066** |
> > > | | | Fortress | 0.110 | 0.087 | 0.106 | 0.089 | **0.093** | **0.087** |
> > > | | **Blender** | Hotdog | **0.246** | 0.515 | 0.250 | 0.516 | 0.247 | **0.513** |
> > > | | | Lego | 0.237 | 0.113 | 0.239 | 0.115 | **0.228** | **0.106** |
> > > | | | Chair | 0.135 | 0.089 | 0.138 | 0.083 | **0.128** | **0.082** |
> > > | | **Mip360** | Bonsai | 0.235 | 0.097 | 0.243 | 0.097 | **0.225** | **0.086** |
> > > | | | Room | 0.251 | 0.119 | 0.249 | **0.100** | **0.236** | 0.102 |
> > > | | | Kitchen | 0.198 | 0.093 | **0.192** | 0.088 | 0.193 | **0.082** |
> > > | | **Mean** | | 0.183 | 0.136 | 0.182 | 0.131 | **0.173** | **0.128** |
> > > | **Long-range** | **LLFF** | Flower | 0.187 | 0.079 | 0.184 | 0.054 | **0.181** | **0.049** |
> > > | | | Horns | 0.312 | 0.203 | 0.307 | 0.185 | **0.307** | **0.182** |
> > > | | | Fortress | 0.206 | **0.201** | 0.208 | 0.202 | **0.193** | 0.204 |
> > > | | **Blender** | Hotdog | 0.378 | 0.652 | 0.385 | 0.655 | **0.377** | **0.648** |
> > > | | | Lego | 0.381 | 0.236 | 0.382 | 0.237 | **0.377** | **0.222** |
> > > | | | Chair | 0.300 | **0.186** | 0.301 | 0.248 | **0.296** | 0.213 |
> > > | | **Mip360** | Bonsai | 0.517 | 0.239 | 0.543 | 0.237 | **0.490** | **0.225** |
> > > | | | Room | 0.578 | 0.296 | 0.587 | 0.271 | **0.568** | **0.270** |
> > > | | | Kitchen | 0.492 | 0.263 | 0.493 | 0.252 | **0.466** | **0.234** |
> > > | | **Mean** | | 0.372 | 0.265 | 0.377 | 0.260 | **0.362** | **0.250** |

---

### Official Review · Reviewer_aRt4 · 2025-06-28

**Clarity:** 3
**Significance:** 3
**Originality:** 3
**Rating:** 4
**Confidence:** 5

**Summary:**

The paper introduces KaRF (Kolmogorov-Arnold Networks-based Radiance Fields), a novel weakly-supervised method for local color editing of neural radiance fields (NeRF).

The first major contribution is the overall KaRF framework architecture. It introduces a two-stage approach that combines segmentation and local recoloring while requiring minimal guiding information for selective recoloring of arbitrary regions. This design achieves multi-view consistency and precise boundary control, addressing key limitations of existing approaches.

On the technical side, the paper makes several innovative contributions. It proposes a novel "residual adaptive gating KAN structure" that enhances segmentation accuracy and editing refinement. The authors introduce a key modification to vanilla KAN by replacing B-spline functions with Gaussian Radial Basis Functions (GRBFs), which better handle high-frequency spatial details and sharp color transitions. They also introduce a palette-based color-adaptive loss to improve color mixing accuracy.

In terms of performance, KaRF demonstrates superior results compared to state-of-the-art methods both qualitatively and quantitatively. The approach shows better handling of color bleeding issues and boundary precision compared to existing approaches.

**Questions:**

Please comment on the limitations raised above.

**Ethical Concerns:**

["NO or VERY MINOR ethics concerns only"]

**Final Justification:**

I acknowledge the authors' detailed responses to my review comments. Their analysis of the GRBF-based KAN modifications adequately addresses one of my primary concerns through experimental validation across multiple scenes and metrics.

The authors have also satisfactorily addressed my concerns regarding computational efficiency and segmentation performance. Their quality-speed tradeoff analysis provides useful insights for practical implementations across different use cases.

Based on these responses, I am satisfied that the authors have addressed the concerns raised in my review and  I maintain my recommendation for acceptance of this paper.

**Limitations:**

Limitations are discussed in the supplementary materials. The reviewer wonders if this is an acceptable practice, as key limitations should be presented in the main paper.

**Paper Formatting Concerns:**

no concerns

**Quality:**

3

**Strengths And Weaknesses:**

Strengths:

Technical Innovation and Soundness:
The paper introduces several novel technical contributions that build meaningfully on prior work. The substitution of B-splines with GRBFs in the KAN framework is well-justified, as it better handles high-frequency spatial details and sharp color transitions. The residual adaptive gating KAN structure shows thoughtful design in addressing training stability while maintaining expressive power.

Clear Problem Motivation:
The authors clearly articulate the limitations of existing NeRF-based editing approaches, particularly around boundary precision, multi-view consistency, and color bleeding. The proposed solution directly addresses these identified gaps in current methods.

Comprehensive Evaluation:
The evaluation is thorough, testing the method across multiple datasets (NeRF-Synthetic, Mip-NeRF 360, LLFF) and comparing against relevant state-of-the-art methods. The inclusion of both qualitative and quantitative metrics, along with user studies, strengthens the validation of their claims.

Limitations

Limited Analysis of KAN Modifications:
One of the paper's most significant contributions is the novel use of KAN for NeRF editing and its modification to incorporate Gaussian RBFs instead of B-splines. However, the analysis of this crucial modification is insufficiently thorough. The main paper only provides one anecdotal comparison in Figure 7, while the supplementary material's Table 6 only shows results for a single scene (kitchen). A more rigorous analysis across multiple scenes and different types of editing operations would be valuable to understand when and why GRBFs outperform B-splines. This would help readers better understand the trade-offs and benefits of this fundamental architectural choice.

Computational Efficiency:
The method requires approximately 10 minutes for processing, which is almost twice as slow as LAENeRF and significantly slower than other recent approaches. For instance, IReNE (Mazzucchelli et al., 2024) achieves similar editing capabilities in about 5 seconds. This performance gap could limit the method's practical applicability, particularly in interactive editing scenarios where quick feedback is essential. The paper would benefit from a discussion of potential optimizations or trade-offs between quality and speed.

While the paper claims superior segmentation performance, the actual improvements over existing methods are relatively modest. As shown in Table 1, KaRF achieves an IoU of 94.7% and accuracy of 99.5%, which represents only a small improvement over SA3D (93.7% IoU, 99.2% accuracy) and MVSeg (90.7% IoU, 98.8% accuracy).

---

> ### Author Rebuttal · Authors · 2025-07-30
>
> Thank you for your detailed review of our work. We have carefully considered your comments and offer our responses below.
>
> **Q1**: Limited Analysis of KAN Modifications.
>
> **Response**: We thank the reviewer for highlighting the importance of thoroughly analyzing our GRBF-based modification to KAN. We agree that this is a central contribution and have conducted additional experiments to provide a more rigorous evaluation.
>
> Expanded Evaluation:
> We systematically compared GRBFKAN with both B-Spline KAN and MLP across six representative scenes (two per dataset) and two key tasks: local recoloring and segmentation.
>
> 1. Local Recoloring:
> As shown in Tab 1, GRBFKAN consistently achieves the best PSNR and SSIM across all scenes. This supports our claim that GRBFs, with their flexible locality and sharper kernel responses, are better at capturing the high-frequency details present in colors and textures than smoother B-splines or global MLPs. Additionally, GRBFKAN exhibits faster and more stable convergence (see supplementary material Figure 7).
>
> **Tab 1**: Impact of the GRBFKAN (PSNR↑ SSIM↑).
> |Scene|MLP|BspineKAN|GRBFKAN|
> |-|-|-|-|
> |LLFF-Horns|35.14 0.971|35.66 0.968|**35.97 0.977**|
> |LLFF-Flower|32.13 0.964|32.94 0.966|**33.52 0.966**|
> |Blender-Chair|33.76 0.986|35.07 0.984|**35.12 0.985**|
> |Blender-Hotdog|36.49 0.967|36.85 0.972|**38.95 0.988**|
> |Mip360-Bonsai|36.96 0.985|36.97 0.977|**38.31 0.989**|
> |Mip360-Kitchen|27.65 0.934|30.29 0.945|**31.63 0.952**|
> |Mean|33.69 0.968|34.63 0.969|**35.58 0.976**|
>
> 2. Segmentation:
> As shown in Tab 2, GRBFKAN consistently outperforms both B-Spline KAN and MLP in average IoU and Acc. This suggests its improved ability to learn geometric and semantic boundaries, which is critical for high-quality editing. Furthermore, the loss convergence plots (see supplementary material Figure 7) reinforce this advantage, showing that GRBFKAN not only converges faster but also achieves a significantly lower final loss.
>
> **Tab 2**: Impact of the GRBFKAN (IoU↑ Acc↑).
> |Scene|MLP|BspineKAN|GRBFKAN|
> |-|-|-|-|
> |LLFF-Horns|0.949 0.992|0.946 0.991|**0.951 0.992**|
> |LLFF-Flower|**0.961 0.991**|0.960 0.991|0.960 0.991|
> |Blender-Chair|0.967 0.997|0.967 0.997|**0.970 0.997**|
> |Blender-Hotdog|0.949 0.994|0.946 0.994|**0.962 0.996**|
> |Mip360-Bonsai|0.862 0.994|0.849 0.993|**0.868 0.994**|
> |Mip360-Kitchen|0.955 0.994|0.952 0.993|**0.961 0.994**|
> |Mean|0.941 0.994|0.937 0.993|**0.945 0.994**|
>
> Summary:
> These results reinforce that GRBFKAN offers clear advantages, especially in challenging scenarios involving complex textures or precise segmentation. We will integrate this extended analysis into the main paper to better justify our architectural choices.
>
> **Q2**: Computational Efficiency.
>
> **Response**: We appreciate the reviewer's concern regarding the processing time. We agree that efficiency is important, especially for interactive scenarios, and would like to elaborate on this from the following perspectives:
>
> 1. Justified Trade-off for Quality:
> KaRF’s relatively longer runtime stems from its use of Kolmogorov-Arnold Networks (KANs), which significantly enhance nonlinear expressiveness. This is critical for achieving superior editing fidelity, multi-view consistency, and fine-grained color control. As shown in Figure 5 (main paper) and Figure 4 (supplementary material), KaRF reduces artifacts like color bleeding, which are often observed in faster methods such as LAENeRF and IReNE.
>
> 2. Favorable Comparison with SOTA Methods:
> While slower than LAENeRF, KaRF is faster than other SOTA editing methods such as PaletteNeRF and the recent 3DGS-based method ICE-G. This positions it as a competitive choice when editing quality is prioritized. Meanwhile, this indicates that KaRF's processing time is within a reasonable range, offering a practical balance between speed and quality.
>
> 3. Quality-Speed Trade-off:
> As shown in Tab 3, We conduct additional experiments to demonstrate tunable speed-quality trade-offs. Reducing the local recoloring stage from 3k to 2k iterations shortens runtime to ~6.5 minutes with only marginal PSNR/SSIM drops. This makes KaRF viable for semi-interactive use cases.
>
> **Tab 3**: Quality-speed trade-off (PSNR↑ SSIM↑).
> |Dataset/Iterations (time)|1k (3.5min)|2k (6.5min)|3k (10min)|
> |-|-|-|-|
> |LLFF|31.49 0.946|32.58 0.961|33.52 0.966|
> |Blender|31.33 0.967|34.42 0.979|35.12 0.985|
> |Mip360|26.15 0.914|29.82 0.939|31.63 0.952|
>
> **Q3**: The improvement in segmentation performance is relatively limited.
>
> **Response**: Thank you for this insightful question regarding our segmentation performance. We agree that when looking solely at the average metrics in Table 1 (main paper), the quantitative gains from KaRF may seem modest.
>
> However, we emphasize that in the context of local color editing, the visual quality and boundary sharpness of the segmentation is far more impactful than marginal gains in overall averages. As shown in Figure 4 (main paper), SA3D often yields blurred or imprecise boundaries, while KaRF produces sharper, texture-preserving masks—essential for preventing color bleeding.
>
> Crucially, segmentation in our framework is not an end goal but a critical intermediate step for enabling high-fidelity local editing. It is this high-precision segmentation that allows KaRF to perform nearly perfect, isolated recoloring. The profound downstream impact of this precision is quantified in Table 2 (main paper): the MSE in non-edited regions for our method is an order of magnitude lower than that of all competing methods. This dramatic reduction in error directly stems from our accurate boundaries, which prevent the color bleeding that plagues other approaches.
>
> In summary, while the metric deltas are small, their downstream impact on editing quality is substantial and meaningful.
>
> **Q4**: Limitations are discussed in the supplementary materials.
>
> **Response**: Thank you for raising this important point. We completely agree with your assessment. In preparing the initial manuscript, due to a strict page limit, we had to make difficult decisions regarding content placement. To dedicate more space in the main paper to our core methodology and experimental results, we moved the discussion on limitations, certain finer experimental details, and additional visualizations to the supplementary material. And we commit to moving the limitations section from the supplementary material back into the main paper. This change will ensure that all readers can easily access our discussion on the scope of the work and potential avenues for future research.

---

### Official Review · Reviewer_TYu2 · 2025-07-01

**Clarity:** 3
**Significance:** 3
**Originality:** 3
**Rating:** 5
**Confidence:** 3

**Summary:**

The paper introduces a new method for palette-based recoloring of NeRFs. The authors introduce the KAN structure to refine the initial semantic segmentations obtained from SAM, obtaining segmentations of high quality. The resulting regions are used to build a hierarchy of masks, palettes and mixing weights following the LoCoPalettes paper. Results show better consistency and user control than previous works.

**Questions:**

The method is sound, presented and evaluated properly. More than a question, it is good to see existing methods for 2D to be repurposed for NeRFs in such a clean and successful way.

**Ethical Concerns:**

["NO or VERY MINOR ethics concerns only"]

**Final Justification:**

I’m keeping my recommendation for acceptance based on my discussion, a the ones the authors had with the other reviewers, who increased their scores after very detailed discussions too

**Limitations:**

yes

**Quality:**

3

**Strengths And Weaknesses:**

- Strengths:
- New method for intuitive recoloring of NeRFs, with results of better quality than previous works.
- Introduction of the KAN network for the mask refinement and recoloring task.
- Clarity of exposition and evaluation.

- Weaknesses:
- Smart combination of existing work, but with great results.
- Processing times seem too high for becoming practical.

---

> ### Author Rebuttal · Authors · 2025-07-30
>
> Thank you for your positive feedback on our paper. We will now provide our responses to the weaknesses and questions you identified.
>
> **Q1**: Smart combination of existing work, but with great results.
>
> **Response**: Thank you for recognizing the strength of our approach. While we build on powerful techniques from multiple domains, our contribution lies in their careful integration, which is tailored for local color editing in neural radiance fields. Specifically, our two-stage KAN-based architecture, with its residual adaptive gating design and palette-based loss, brings notable improvements in segmentation precision and local recoloring quality. We appreciate your positive feedback on our results.
>
> **Q2**: Processing times seem too high for becoming practical.
>
> **Response**: Thank you for the reviewer’s insightful comment regarding the processing time. We agree that efficiency is crucial for practical applications, and we address this concern from the following perspectives:
>
> 1. Trade-off Between Quality and Speed:
> KaRF is designed to prioritize high-fidelity editing, multi-view consistency, and fine-grained control. This naturally incurs higher computational cost compared to some faster methods (e.g., LAENeRF, IReNe), but yields superior editing quality, as shown in Figure 5 (main paper) and Figure 4 (supplementary material). We believe this trade-off is justified, especially for applications demanding precision.
>
> 2. Source of Computational Overhead:
> The primary cost arises from using Kolmogorov-Arnold Networks (KANs), which offer significantly greater nonlinear expressiveness than MLPs. As validated in our ablation studies, this contributes directly to better segmentation and local recoloring performance, albeit with added computation.
>
> 3. Reasonable Compared to SOTA Methods:
> While our method takes longer than LAENeRF, its processing time is favorable when compared to other SOTA editing methods. KaRF is faster than both PaletteNeRF and the recent 3DGS-based method ICE-G. This indicates that the processing time of KaRF is within a reasonable range, offering a practical balance between speed and quality.
>
> 4. Potential for Acceleration:
> KaRF is flexible and allows for quality-speed trade-offs. By reducing the local recoloring iterations from 3k to 2k or 1k, we can significantly reduce runtime (down to ~6.5 min) while maintaining respectable quality, as shown in Tab 1. This suggests good potential for deployment in semi-interactive scenarios.
>
> **Tab 1**: Quality-speed trade-off (PSNR↑ SSIM↑).
> |Dataset/Iterations (time)|1k (3.5min)|2k (6.5min)|3k (10min)|
> |-|-|-|-|
> |LLFF|31.49 0.946|32.58 0.961|33.52 0.966|
> |Blender|31.33 0.967|34.42 0.979|35.12 0.985|
> |Mip360|26.15 0.914|29.82 0.939|31.63 0.952|
>
> **Q3**: The method is sound, presented and evaluated properly. More than a question, it is good to see existing methods for 2D to be repurposed for NeRFs in such a clean and successful way.
>
> **Response**: We are truly grateful for such a positive and encouraging assessment of our work. We are delighted that you found our method to be sound and presented and evaluated properly. Your observation perfectly captures our core motivation: to repurpose powerful, existing 2D methods for NeRFs in a clean and successful way. We hope our method can build a bridge between mature 2D technologies and emerging 3D representations.

---

> > ### Comment · Reviewer_TYu2 · 2025-08-04
> >
> > Thank you for your replies

---

> > > ### Author Response · Authors · 2025-08-07
> > >
> > > My pleasure. Thank you for your valuable time and constructive feedback!

---

### Note · Authors · 2025-08-11

We would like to express our sincere gratitude to the reviewers for their constructive feedback and to the AC for your valuable time. We have provided detailed and faithful responses to each of the weaknesses and questions raised. We deeply appreciate your support.

Please note that due to the limitations of the rebuttal format, we were unable to include images. In their place, we have provided ample quantitative results and described some of the key visual outcomes. We assure you that, should the paper be accepted, we will incorporate more comprehensive visual experiments into the main paper and the supplementary materials.

---

### Decision · Program_Chairs · 2025-09-17

**Decision:**

Accept (poster)

**Comment:**

The paper introduces KaRF, a framework for local color editing of NeRFs using Kolmogorov-Arnold Networks (KANs). Key contributions include a residual adaptive gating KAN with GRBF activations and a palette-based loss, enabling improved boundary precision and reduced color bleeding. Reviewers highlighted the novelty of applying KANs to NeRF editing, the clear exposition, and strong results across benchmarks. Main concerns focused on the supervision level (multiple SAM masks, stronger than “weak”), limited ablations and direct MLP comparisons, slow runtimes (~10 min vs seconds for recent methods), and insufficient qualitative evaluation under complex lighting or 360° settings.

The rebuttal included additional ablations and clarified quality–speed trade-offs, addressing some architectural and efficiency concerns. However, questions remain about scalability, supervision strength, and evaluation breadth. Overall, I find the paper technically solid and innovative, with contributions of interest to the community. I recommend acceptance, as the strengths outweigh the weaknesses but stronger empirical validation would improve confidence in the claims. The Camera Ready version should include additional results and clarifications provided in the rebuttal.